

# How waviness in the circulation changes surface ozone: a viewpoint using local finite-amplitude wave activity

Wenxiu Sun[1,*], Peter Hess[1], Gang Chen[2], and Simone Tilmes[3]

[1]Department of Biological and Environmental Engineering, Cornell University, Ithaca NY, USA
[2]Department of Atmospheric and Oceanic Sciences, University of California, Los Angeles, Los Angeles CA, USA
[3]National Center for Atmospheric Research, Boulder CO, USA
[*]Currently at BloomSky Inc., Burlingame CA, USA

*Correspondence to:* Wenxiu Sun (ws299@cornell.edu)

**Abstract.** Local finite-amplitude wave activity (LWA) measures the waviness of the local flow. In this work we relate the anticyclonic part of LWA, AWA (Anticyclonic Wave Activity), to surface ozone in summertime over the U.S. on interannual to decadal scales. Interannual covariance between AWA diagnosed from the European Centre for Medium-Range Weather Forecast Era-Interim reanalysis and ozone measured at EPA Clean Air Status and Trends Network (CASTNET) stations are analyzed using Maximum Covariance Analysis (MCA). The first two modes in the MCA analysis explain 84% of the covariance between the AWA and MDA8 (Maximum Daily 8h-Average ozone). Over most of the U.S. we find a significant relationship between ozone at any specific location and AWA over the analysis domain (24°N-53°N, and 130°W-65°W) using a linear regression model. This relationship is diagnosed (i) using reanalysis meteorology and measured ozone from CASTNET, or (ii) using meteorology and ozone simulated by the Community Atmospheric Model version 4 with chemistry (CAM4-chem) within the Community Earth System Model (CESM1). Using the linear regression model we find that meteorological biases in AWA in CAM4-chem, as compared to the reanalysis meteorology, induces ozone changes between -4 and +8 ppb in CAM4-chem. Future changes (circa 2100) in AWA are diagnosed in four different climate change simulations in CAM4-chem, simulations which differ in their initial conditions and in one case in their reactive species emissions. All future simulations have enhanced AWA over the U.S., with the maximum enhancement in the southwest. As diagnosed using the linear regression model the future change in AWA is predicted to cause a corresponding change in ozone ranging up to ± 6 ppb. The location of this change depends on subtle features of the change in AWA. In many locations this change explains the magnitude and the sign of the overall simulated future ozone change.

## 1 Introduction

Tropospheric ozone impacts human health (McKee, 1993), the environment (e.g., Arneth et al., 2010), and climate (IPCC, 2013). The purpose of this study is to examine whether local wave activity (LWA), a newly developed diagnostic of the waviness of the atmospheric flow, can be used over the continental US to: 1) quantify present day surface ozone variability and 2) predict the extent that future changes in atmospheric circulation will impact the surface ozone concentration.



Previous studies have shown that surface ozone is correlated with local meteorological factors such as surface temperature (e.g., Brown-Steiner et al., 2015), frontal passages (e.g., Ordónez et al., 2005) and stagnation (e.g., Jacob and Winner, 2009; Sun et al., 2017), although Kerr and Waugh (2018) show only a weak relationship on daily timescales between ozone and stagnation. In many regions, surface temperature is the largest covariate of surface ozone (Porter et al., 2015; Oswald et al.,

2015). However, in the northeast US extended stagnation episodes predict high ozone events better than temperature alone (Sun et al., 2017). Shen et al. (2015) note the importance of both the local and regional meteorological scales (e.g., synoptic scale circulations) in determining ozone variability. On the larger scales regional scales jet position Barnes and Fiore (2013), the 500 hPa geopotential height (Lin et al., 2014; Shen et al., 2015), the Bermuda high location (Shen et al., 2015) and the frequency of cyclone passages (e.g., Leibensperger et al., 2008) all have been shown to impact local meteorological conditions and ozone.

Less is known about the relation between ozone and features of the general circulation (but see Young et al. (2017)) although ozone has been related to various indexes of the circulation including the Pacific Decadal Oscillation Oswald et al. (2015), the Quasi-Biennial Oscillation Oswald et al. (2015), El Nino and Southern Oscillation (e.g., Shen and Mickley, 2017; Xu et al., 2017), the Arctic Oscillation (e.g., Oswald et al., 2015; Hess and Lamarque, 2007), and variations in stratosphere-troposphere exchange (e.g., Hess and Zbinden, 2013; Hess et al., 2015). While large scale variables do not outperform the local variables

in terms of their predictive power for ozone (Oswald et al., 2015) the impact of climate change on large scale features of the circulation is likely more robust than that on smaller scales and in some cases, large-scale changes in circulation can be inferred from general theoretical arguments.

In this study we explore a general way to explain ozone's variability in terms of large scale synoptic conditions through the

local wave activity (LWA) of the mid-tropospheric flow. Derived from the divergence theorem, finite-amplitude wave activity (Nakamura and Zhu, 2010) mathematically relates large scale wave dynamics to the atmospheric circulation (Nakamura and Solomon, 2011; Methven, 2013; Chen and Plumb, 2014; Lu et al., 2015). LWA generalizes the zonally averaged finite amplitude wave activity to local longitudinally dependent scales. It can be used to differentiate longitudinally isolated events and to characterize local and regional weather (Huang and Nakamura, 2016). A diagnosis of LWA also provides a metric for the

occurrence of blocking events, events associated with anomalous or extreme mid-latitude weather events such as heat waves (Chen et al., 2015; Martineau et al., 2017) which have been associated with surface ozone extremes (e.g., Sun et al., 2017; Meehl et al., 2018; Phalitnonkiat et al., 2018). Since blocking events are related to the flux and convergence of LWA, the processes that control LWA may provide clues to how blocking will change in the future (Nakamura and Huang, 2018). Thus LWA potentially makes a good candidate for relating surface ozone to characteristics of the general circulation.

Climate change causes notable and well documented changes on surface ozone through changes in both chemistry and changes in circulation. Changes in atmospheric chemistry from increased temperature and water vapor can either increase or decrease surface ozone, dependent on surface emissions. An increase in the strength of the Brewer Dobson circulation is a robust feature of future simulations (Garcia and Randel, 2008), although it is unclear as to the extent to which the associated

increase in the stratosphere-troposphere exchange of ozone extends to the surface (Collins et al., 2003). Evidence of zonally



symmetric changes in the future mid-latitude circulation (e.g., the increase in the strength of the Brewer Dobson circulation) are more robust than that of regional changes. However, the use of LWA to diagnose circulation changes emphasizes zonally asymmetric changes with associated regional impacts.

There is some evidence that a climate-induced shift in storm tracks will be zonally asymmetric. The CMIP5 models predict a poleward shift in the jet position in the North Atlantic (Barnes and Polvani, 2013) although the Pacific storm track shows little movement with climate change (Shaw et al., 2016). The northwards shift in the North Atlantic jet will likely decrease ozone variability over the Northeast US and change the relationship between temperature and ozone (Barnes and Fiore, 2013). These changes in the storm-track may also be related to changes in summertime cyclone frequency, reported in some (e.g.,
Turner et al., 2013) but not all studies (Lang and Waugh, 2011) over the U.S. However, note that Turner et al. (2013) show the relationship between cyclone frequency and high-ozone events is weak. Over Europe notable future changes are predicted (Masato et al., 2013) in summertime blocking events. These events with their accompanying atmospheric persistence and temperature extremes are also associated with pollution extremes. However, blocking events are rare over the US during the summer months. However, the North Atlantic subtropical anticyclone (commonly referred to as the Bermuda High, (Davis
et al., 1997)) has shown a consistent tendency in future model simulations to intensify and move to the west (e.g., Li et al., 2012; Shaw and Voigt, 2015). Ozone over the eastern part of the US is sensitive to the position and the variability of the Atlantic subtropical anticyclone (Shen et al., 2015). Horton et al. (2014) predicts an increase in air stagnation over the southwestern US in the future, consistent with an increase in the future anticyclonic circulation in the southwestern part of the country (Shaw and Voigt, 2015). However, on the whole relatively little is known about how future zonally asymmetric circulation changes
will impact surface ozone on a regional scale.

Here we focus on the extent that LWA is related to surface ozone and use this to predict the impact of circulation changes on ozone in the present and future climates. Section 2 introduces the data and the methods. In Section 3 we 1) explore the relationship between ozone and LWA in the present climate in model simulations and observations and 2) apply a simple uni-
variate linear regression model to predict the ozone change in the future due to the change in LWA. Discussion is in section 4 with conclusions in section 5.





## 2   Data and Method

### 2.1   Local wave activity calculation

The wave activity is calculated following Chen et al. (2015). To calculate LWA, for a quantity q (in this study we use geopotential height at 500 hPa, $Z_{500}$) decreasing with latitude in the northern hemisphere, we first define the equivalent latitude $\phi_e(Q)$
(Eq. 1), in which S(Q) is the area bounded by the q=Q contour towards the north pole (see Figure 1):

$$\phi_e(Q) = \arcsin[1 - \frac{S(Q)}{2\pi a^2}] \tag{1}$$

The area within the equivalent latitude circle is equal to the area within the q=Q contour. The cyclonic (southern) and anticyclonic (northern) LWA at longitude $\lambda$ and latitude $\phi_e$ can be defined as

$$A_C(\lambda,\phi_e) = \frac{a}{cos\phi_e} \int_{\hat{q}\leq 0,\phi\leq\phi_e(Q),\lambda=const} \hat{q}cos\phi d\phi \qquad A_A(\lambda,\phi_e) = \frac{a}{cos\phi_e} \int_{\hat{q}\geq 0,\phi\geq\phi_e(Q),\lambda=const} \hat{q}cos\phi d\phi \tag{2}$$

where $\hat{q} = q - Q$ (See Fig. 1). Thus, the cyclonic and anticyclonic LWA is defined by integrating the eddy term ($\hat{q}$) southward and northward, respectively. The total LWA ($A_T$) is defined as in Huang and Nakamura (2016)

$$A_T(\lambda,\phi_e) = A_C - A_A \tag{3}$$

As defined in Eq. 2, $A_C \leq 0$ in the northern hemisphere, representing the cyclonic wave activity to the south of the equivalent latitude $\phi_e$, and $A_A \geq 0$, representing the anticyclonic wave activity to the north. This formulation can be used to quantify
the waviness in the time mean field as in Fig. 1 as well as the daily field with synoptic variability. More details on LWA theory and derivation can be found in Chen et al. (2015) and Huang and Nakamura (2016). Because the anticyclonic wave activity contributes to most of the total LWA over the continental U.S. (Fig. 1, Fig. 2) for simplicity and significance we relate the ozone concentration to anticyclonic wave activity (AWA) in the results below (the use of AWA gives similar results to using LWA). In addition, AWA is associated with high pressure systems which oftentimes relate to high ozone events.

### 2.2   Measured and Simulated data

The extent to which LWA can explain surface ozone variations is examined during the present day in both measured and simulated data during the summer months. We analyze the measured and simulated relationship between LWA and surface ozone
concentrations in a study region defined as the region between 24°N-53°N, and 130°W-65°W (see Fig.2). This region, covering the continental US, is large enough to quantify the impact of non-local changes in LWA on surface ozone over the US, but not so large as to measure the impact of more distant teleconnection patterns. For the measured relationship we relate surface ozone concentrations measured at EPA Clean Air Status and Trends Network (CASTNET) sites to the AWA derived from meteorological analysis. The simulated relationship between LWA and ozone is derived using the Community Atmospheric Model



version 4 with chemistry (CAM4-chem) of the Community Earth System Model (CESM1). The simulated future changes in AWA (circa 2100) are then used to predict future changes in ozone due to future circulation change.

The measured relationship between LWA and ozone is analyzed for the summer months (June, July, August, JJA hereafter) from 1994-2013. The measured surface ozone is taken from 49 CASTNET sites (see Fig. 3). At many measurement sites a decreasing trend is found in ozone during this time period (e.g., Cooper et al., 2014). Therefore, we normalize the ozone field by splitting the study period into two equal length periods, remove the 21-day smoothed JJA seasonal cycle, and normalize the ozone anomalies for each sub-period (Phalitnonkiat et al., 2018). The procedure produces a quasi-stationary time series for ozone at each site (Sun et al., 2017). In all cases we relate the Maximum Daily 8h-Average ozone (MDA8 hereafter) to the wave activity. When analyzing the measured relationship between ozone and LWA, 500 hPa geopotential height from European Centre for Medium-Range Weather Forecasts's Era-Interim dataset (Dee et al., 2011) is used to compute LWA over the study region for the period of 1994-2013.

The Community Atmospheric Model version 4 (CAM4-chem) of the Community Earth System Model (CESM1) is used to simulate the relationship between ozone and LWA over the study region. CAM4-chem is described in Lamarque et al. (2012). The simulated trend and magnitude of surface ozone in CAM4-chem has been largely improved compared to the earlier versions of the model because of the updates in the chemistry scheme, dry deposition rates and radiation and optics due to a new treatment of aerosols (Tilmes et al., 2016). The resolution in the simulations analyzed below is $1.9°$ latitude $\times$ $2.5°$ longitude, with a model top of approximately $\sim 40$ km.

Three ensemble members of the CESM1 CAM4-chem (REFC2) ensemble members were run from 1960-2100 using the Climate Chemistry-Climate Model Initiative (CCMI) procotol. This protocol follows the RCP 6 (Representation Concentration Pathway 6) scenerio (Tilmes et al., 2016) pathway for ozone and aerosol precursor emissions and mixing ratios for carbon dioxide, methane and nitrous oxide (Table 1). CCMI is coordinated by the International Global Atmospheric Chemisty/Stratospheric Processes for the evaluation and intercomparison of chemistry-climate models, with the participation of many fully coupled chemistry-climate models (Eyring et al., 2013).

In addition to these three ensemble simulations we perform two climate-change-only simulations in order to evaluate isolate the impact of climate change on ozone (see Table 1): a present day (the GCM2000 simulation) and a future simulation (the GCM2100 simulation). These simulations are branched off of the first ensemble member of the CESM1 CAM4-chem REFC2 simulation, one in 2000 and one in 2100 respectively. In both the GCM2000 and GCM2100 simulations the emissions of all species including biogenic emissions and the concentrations of long-lived species (including CH4) are kept constant at the 2000 level. In the present day simulation, the atmospheric CO2 was set at 369 ppm, representative of conditions circa 2000, while in the future simulation CO2 was set at 669 ppm, representative of conditions in 2100 following the RCP 6 scenario. In the GCM2000 and GCM2100 simulations the simulation period is 26 years, with the latter 20 years (2006-2025 or 2106-2125)





used for the analysis. The average temperature change over the continental U.S. between the GCM2100 simulation and the GCM2000 simulation is 2.1°C, smaller than the 2.8°C computed using the parent CCMI REFC2 simulations. This is likely due to the fact the aerosol emissions are held constant at the 2000 levels in both the present and future climate simulations in the GCM2100 and GCM2000 simulations. The simulations are summarized in Table 1. Further analysis of these simulations

is given in Phalitnonkiat et al. (2018). In the analysis both the MDA8 ozone and LWA time series are detrended by removing the 21-day smoothed JJA seasonal cycle.

## 2.3   Maximum Covariance Analysis

Maximum Covariance Analysis (MCA) (Wilks, 2011) finds the patterns in two spatially and temporally varying fields that

explains the maximum fraction of covariance between them. We use MCA to find the overall relationship between ozone and LWA over the study region. Explicitly, the steps calculating the MCA are as follows: 1) compute the covariance matrix for ozone and AWA over the considered domain; 2) perform Singular Value Decomposition (SVD) on the covariance matrix and obtain the modes that both dominate the ozone and AWA time series and strongly correlate with one another (resulting in the maximum covariance between the two time series).

## 2.4   The univariate linear regression model

A univariate linear regression model is used to quantify the relationship between ozone and AWA at individual points within the study region. The linear relationship between changes (in time) of normalized ozone at a point $(i_0, j_0)$ and changes in normalized wave activity at another point $(i, j)$, can be simply expressed as the slope of ozone with respect to wave activity

$(ppb/m^2)(S_{i_0,j_0(i,j)})$.

Summing over all points within the study region gives the projection (denoted by $p_{i_0,j_0}$ (ppb)) of AWA on ozone$(i_0, j_0)$. Thus $p_{i_0,j_0}$ is calculated as:

$$p_{i_0,j_0} = \boldsymbol{AWA} \cdot \boldsymbol{S}_{i_0,j_0} = \sum_j \sum_i AWA(i,j) \times S_{i_0,j_0}(i,j) \tag{4}$$

The advantage of calculating the projection value $p$ is that it measures in a single variable the similarity between the AWA and the ozone sensitivity to AWA, incorporating the regional impact of AWA on ozone. However, the projection in Eq. 4 can result in an over prediction of the contribution of AWA to ozone as the summation in Eq. 4 is not over statistically independent





points. Therefore, to correct for this we build a linear regression model where we relate $O_{3i_0,j_0}(t)$ to $p_{i_0,j_0}(t)$ through the linear regression coefficients $\alpha_{i_0,j_0}$ and $\beta_{i_0,j_0}$:

$$O_{3i_0,j_0} = \beta_{i_0,j_0} \times p_{i_0,j_0} + \alpha_{i_0,j_0} \tag{5}$$

On the daily timescale the relation between ozone and the projection value is quite noisy. Therefore, to reduce the noise we
only apply Eq. 5 on an interannual timescale. Specifically we relate the interannual changes in ozone averaged over JJA to interannual changes in the JJA projection value. In turn the JJA averaged projection value ($p$) is calculated from the JJA averaged AWA multiplied by the slope ($S$) (Eq. 4). However, the slope ($S$) is noisy when calculated from interannual variations as we only have 20 years of data at our disposal. Thus we calculate $S$ using daily variations of ozone and wave activity. In summary, the projection value for each year (Eq. 4) is calculated using the JJA averaged wave activity for each year, but with
$S$ determined from daily variations.

In our analysis below we ascribe the change in ozone due to the change in AWA as:

$$\Delta O_3 = \beta[(\Delta \boldsymbol{AWA}) \cdot \boldsymbol{S}] \tag{6}$$

We apply this equation to: (i) calculate the ozone bias that can be traced to differences in the AWA between the GCM2000 simulation and the ERA-Interim reanalysis; (ii) calculate the ozone change that can be traced to future changes in the AWA. In both these cases we compute the climatological change in AWA, project it onto S, and multiply it by the slope $\beta$. We assume that $S$ and $\beta$ remain constant. For example, in the future we assume ozone responds to changes in the future circulation as it would respond to those same circulation changes in the present climate. Note that $\beta$ (and $S$) can be derived from the GCM2000 simulation at all points or from the measurements at CASTNET sites. We will refer to these different estimates of $\Delta O_3$ as the simulation-derived, or measurement-derived $\Delta O_3$.

## 3   Results

The climatological wave-activity over the study region in the ERA-Interim reanalysis and in the GCM2000 simulation is given in Sect. 3.1, while the relation between variability in AWA and ozone is given in Sect. 3.2 using MCA. Sect. 3.3 analyses the extent to which we can explain interannual changes in ozone from changes in AWA using the univariate linear regression model. This model is then used to explain the extent to which AWA differences between the GCM2000 simulation and the ERA-Interim reanalysis result in differences in the ozone distribution (Sect. 3.4) and the extent to which AWA differences
between the GCM2100 simulation and the GCM2000 simulation result in ozone differences (Sect. 3.5).





## 3.1 Climatological Mean Wave Activity

Positive AWA anomalies are associated with 500 hPa ridges and CWA anomalies with 500 hPa troughs (Fig. 2). The mean JJA
LWA over the US in both the ERA-Interim reanalysis and the GCM2000 simulation is dominated by the anti-cyclonic compo-
nent. In both datasets AWA anomalies are centered over the southwestern US with cyclonic wave activity mostly confined to

the northeast and northwest of the study region. In each dataset LWA (Fig. 2, last row) and AWA (Fig. 2, first row) are similar
over the domain. While the GCM2000 simulation has a very dominant AWA maximum centered over the southwestern US the
ERA-Interim analysis has two maxima of approximately equal strength: one centered over the southwestern US and the other
centered near 30-35$^o$N on the eastern border of the domain. The southwestern AWA maxima in the ERA-Interim analysis is
much weaker than in the GCM2000 analysis. The simulated and reanalyzed LWA differences could be due to model bias or to

due internal variability (e.g., Deser et al., 2012) in the GCM2000 simulation.

## 3.2 Maximum Covariance Analysis

MCA is used to examine the overall pattern of covariance between ozone and AWA (Fig. 3) as derived from measurements and
from the GCM2000 simulation. In one case ozone is taken from the measurements at CASTNET sites while AWA is calcu-

lated from ERA-Interim analysis; in the other case both ozone and AWA are taken from the GCM2000 simulation, where ozone
is sampled at the CASTNET sites. This analysis emphasizes the Eastern US due to the high density of CASTNET stations there.

The first two modes identified in the MCA analysis derived from measurements (Fig. 3 a,b) explain 84% of their covariance,
with the first mode explaining 58% of the covariance, followed by 26% explained by the second mode (Fig. 3). The first mode

consists of a positive anomaly of AWA off the east coast of the U.S. and Canada, a negative AWA anomaly centered southwest
of the Great Lakes, and a positive AWA anomaly in the western portion of the US. In the eastern third of the country the ozone
anomaly is negative (up to -12 ppb). In the second mode a strong positive AWA anomaly is located on the eastern coast of the
US, with a negative AWA anomaly over the center of the country. In the second mode the ozone anomalies range from strongly
positive over the northeast U.S. (up to 9 ppb) to negative over the southeast U.S. (up to -10 ppb). If one uses the 500 hPa (Z500)

geopotential heights instead of AWA this analysis (Fig. S1) one obtains very similar results with small displacements in the
Z500 anomaly compared with the AWA anomaly.

The ozone anomalies identified here in the first two MCA modes are consistent with the leading two empirical orthogonal
functions (EOFs) of ozone over the eastern U.S. identified in Shen et al. (2015) . Similar to our results, the first ozone EOF

identified in Shen et al. (2015) has a negative ozone anomaly throughout the eastern US with the largest negative anomalies
near 45$^o$ N while the second EOF has a positive ozone anomaly over the northeast US with negative anomalies over the south-
east US and the Gulf Coast. Moreover, the correlation between the first two ozone EOFs and geopotential height in Shen et al.
(2015) is consistent with the wave activity (or geopotential height) identified here in the first two modes of the MCA analysis.





It is worth stressing that while the results in Shen et al. (2015) were obtained by first finding the leading EOFs of ozone, then correlating these EOFs with the geopotential height, the MCA methodology picks out the ozone and AWA (or geopotential height) anomalies in one procedure. Shen et al. (2015) attribute the first mode of the MCA to the impact of low pressure systems crossing the eastern U.S., and associate the second with the westward expansion of the North Atlantic subtropical cyclone. A

westward expansion of this anticylone results in a negative ozone anomaly in the Gulf states as ozone depleted air is advected inland from the gulf, but a positive ozone anomaly in the northeastern U.S. due to the advection of polluted mid-western around the anticyclone to the southeast.

Similar to the results derived from measurements, the first simulated MCA mode explains 66% of the total covariance,
with the second mode explaining 24% (Fig. 3 c,d). The AWA in the first simulated mode differs only slightly from that in the reanalysis. The simulated AWA negative anomaly over the continental U.S. is displaced slightly to the west of that from the ERA-Interim reanalysis. As a result, in the simulation the northeastern US is not subject to the cyclonic flow associated with the first observed mode, and consequently, the simulated ozone anomalies over the very northeastern US are positive, not negative as in the observations. The second simulated mode differs more substantially from that observed with both the positive
and negative AWA anomalies displaced substantially to the west (Fig. 3 b,d)), with weak positive ozone anomalies extending from the northeast US south to Florida along the Atlantic seaboard. In contrast to the observations, the positive AWA anomaly in the second simulated mode is not correctly placed to advect high pollutant concentrations into the northeast US from the Ohio valley. In addition, the simulated negative ozone anomalies in the southeastern US attributed to transport of low ozone air from the Gulf are less also extensive than measured.

The discrepancy between the observationally based and simulated MCA modes may, at least in part, stem from the simulation of the North Atlantic subtropical cyclone (Fig. 4). In the ERA-Interim reanalysis the center of this anticyclone is located to the southeast of the simulated position, the center is broader than in the simulation, and the simulated low pressure trough along the southeastern coast is not evident (Fig. 4). On the other hand, the western extension of the Atlantic anticyclone into the
southeastern US in the reanalysis and the simulation are similar. This suggests that the longitudinal variability of the position of this anticyclone is greater in the reanalysis than in the simulation, most likely leading to a larger range of impacts on continental ozone. Consistent with these differences the MCA modes derived from the observations have rather strong associated ozone anomalies in the northeast US (strongly negative in the first mode and strongly positive in the second mode) while in the simulation these ozone variations are notably weaker. Phalitnonkiat et al. (2018) attributes the rather poor simulation of the
temperature-ozone correlation in the northeastern U.S. to the poor simulation of the Atlantic anticyclone while Zhu and Liang (2013) notes deficiencies, in general, in the ability of general circulation models to simulate this subtropical high.





### 3.3 Univariate Regression Analysis

At any point, the overall change in ozone attributed to changes in AWA is proportional to the change in AWA projected onto the regression coefficients (ppb/m$^2$) calculated for that point (Eq. 6). The regression coefficients (ppb/m$^2$) obtained between the ERA Interim reanalysis (GCM2000 simulation) and measured ozone (simulated ozone) show strong, non-local and significant

relationships between AWA and ozone at all sites examined. For all sites the regression coefficient is positive at the site itself. The measurement derived and simulation derived regression coefficients are in general agreement, although some differences in magnitude and location do occur. We return to these regression coefficients when we examine future ozone changes in Sect. 3.3.2.

At most CASTNET sites and throughout most of the simulated domain interannual changes in AWA explain a significant fraction of the interannual MDA8 ozone variability as determined by the linear regression equation (Eq. 5) (Fig. 5). Changes in AWA explain very little of the simulated ozone variability along the west coast of the US and over the northeast US. In the western US it is possible that changes in AWA outside the study region would hold more explanatory power. Note, that in contrast to the simulation, the measurement derived regression analysis explains a considerable fraction of the measured ozone

variability over the northeastern US. A likely explanation is that over the northeast US the simulated variability (as captured by the MCA modes) is smaller than that measured and thus likely harder to capture.

It is equally possible to build a regression equation based simply on geopotential heights instead of wave activity. The regression coefficients based on the simulated and measured relationships between ozone and geopotential heights within the study

area tend to be somewhat higher than those based on AWA (Fig. S3a), notably in the northeast US. However, it is important to note that wave activity is reflective of asymmetric regional circulation changes with respect to the zonal mean. In contrast, the index based solely on geopotential height is also sensitive to zonally symmetric changes and so is sensitive to a general northward or southward displacement of the jet. In addition, geopotential height can be affected by a uniform change in temperature (equivalently geopotential thickness) that may be unrelated to regional circulation changes. We return to this point in Sect. 3.3.2.

### 3.3.1 AWA differences between GCM2000 and ERA-Interim Reanalysis: Ozone Impacts

Differences in the simulation of AWA between the GCM2000 and the ERA-interim analysis are substantial (Fig. 6a). Over the southwestern US the GCM2000 simulation substantially over-predicts the AWA compared to the ERA-Interim reanalysis; over the Eastern seaboard the GCM2000 simulation underestimates the AWA due to the anomalous cyclonic flow over the coast.

The change in ozone that can be attributed to these simulation differences in AWA results in approximately a 5-10 ppb ozone increase in the interior southeastern US and a decrease of up to 5 ppb in the northeast. Similar to many GCMs ozone is biased high in the GCM2000 with positive biases in all regions of the country including the northeastern US ($\sim$21 ppb), southeastern US ($\sim$20 ppb) and the midwestern regions ($\sim$23 ppb) (Phalitnonkiat et al., 2018). Differences in the climatological wave





activity between the GCM2000 simulation and the ERA-interim analysis acts to decrease the ozone bias over the northeastern states. If the wave activity was unbiased ozone would even be higher in the northeastern US in the GCM2000 simulation. The difference in climatological wave activity between the GCM2000 simulation and the ERA-interim analysis increases ozone in the mid-Atlantic and southeastern states (Fig. 6b).

Ozone in the northeastern US is particularly sensitive to AWA over the eastern seaboard (Sect. 3.3.2). The ozone decrease over the northeast US in Fig. 6 can be attributed to decreased anti-cyclonic activity over the eastern seaboard in the GCM2000 simulation, which acts to decrease the advection of pollutants into the northeast. From the mid-Atlantic states to the southeastern states ozone is particularly sensitive to AWA to the west and southwest of a particular site (Sect. 3.3.2). Ozone differences in these regions are impacted by the stronger ridging in the southwest US in the GCM2000 simulation relative to the ERA-Interim analysis. The positive ozone anomaly in the southeastern US in Fig. 6 can thus be attributed to the relatively large anticyclonic activity in the GCM2000 simulation in the southwestern US which acts to advect continental air with relatively high pollutants concentrations into the southeastern US.

### 3.3.2   Future AWA Changes: Ozone Impacts

Future changes in JJA 500hPa geopotential height and wave activity between the GCM2100 simulation and the GCM2000 simulation are given in Fig. 7. Significant increases in geopotential height occur everywhere, with increases of approximately 30-60 m over most of the U.S. This increase can be attributed to mid-to-high latitude warming in the future climate. The geopotential height increase tends to be larger at higher latitudes consistent with other model projections (e.g., Yue et al., 2015; Vavrus et al., 2017). By definition the zonally symmetric change in geopotential height has no change in AWA, but regional changes in AWA reflect future changes in the waviness of the flow. The most pronounced future change is the large anticyclonic wave activity enhancement over the Southwestern US (Fig. 7b), also seen in the total wave activity (Fig. 7d), in response to increased ridging in this region. Using a different metric to characterize the waviness of the circulation Vavrus et al. (2017) found a large increase in the waviness of the flow (measured as sinuosity) over the U.S. centered at 42°N. In contrast the cyclonic wave activity (CWA) shows relatively small changes in the future (Fig. 7d).

All future ensembles examined show a similar pattern in the future change in AWA (Fig. 8) with larges increases in AWA in the western and southwestern US. Here we examine both the AWA change in the climate simulations (GCM2100 minus GCM2000) where forcing from short-lived constituents and methane remains constant over the 21st century, and the AWA change from each of the three ensemble members of the REFC2 simulation, where the forcing from the short-lived constituents and methane follows the RCP6 scenario. Note, that these future changes in AWA are similar to present day differences between the GCM2000 simulation and the ERA-Interim reanalysis (Fig. 6). Despite the similar pattern in future AWA change in all simulations there are also some significant differences in the strength, position and orientation of the AWA anomaly in the western US. These variations are evident even in a 25 year average, variability attributed to the substantial internal variability



of long-term tropospheric flow (e.g., Deser et al., 2012). As we show below these differences result in substantial uncertainty as to the future ozone change due to changes in regional circulation.

To predict the future change in ozone due to AWA changes we use the regression between ozone and AWA determined in
the present climate. To check that this relationship does not change in the future we calculate the slopes of the linear regression model using AWA and ozone from the present climate (GCM2000) and future climate (GCM2100), and then we construct the 95% confidence intervals for the two slopes. For all the grid points, the 95% confidence intervals of the two slopes are overlap. Therefore, we assume the regression between AWA and ozone does not substantially change in the future.

Regional ozone changes attributed to the future changes in AWA range from approximately -2.5 ppb to 2.5 ppb (Fig. 9). The predicted future changes are smaller than those derived from present day differences between the GCM2000 simulation and the measurement-derived ERA-Interim analysis (Fig. 6), but show an overall similar pattern. All future simulations show an increase in ozone over a portion of the southeastern US, although the amplitude and extent of the increase varies from simulation to simulation. The ozone change over the northeast US is inconsistent between the different simulations with slight ozone
increases or decreases simulated. Over the Rocky Mountains ozone decreases are predicted when future changes are calculated with the simulated regression coefficients, but ozone increases are predicted when calculated with measurement derived coefficients. The variation in ozone change between the different ensembles is consistent with the unforced, low-frequency climate induced variability in ozone as analyzed in (Barnes et al., 2016).

In many locations the change in ozone predicted from the change in wave activity between the GCM2100 and GCM2000 simulations is consistent with the actual ozone change. The ozone increase in the southeastern US through the Pacific Northwest and the ozone decrease in the interior southwest predicted by the linear regression model (in particular the regression based on the GCM2000 simulation) (see Fig. 9) largely agrees in sign with the ozone difference between the GCM2100 simulation and the GCM2000 simulation (Fig. 10). In many of these locations the linear regression also explains the magnitude of the
simulated ozone change. Overall, the RMSE of the linear regression model compared with the GCM simulations is $\sim$ 2 ppb. In the northeastern US the predicted ozone change from the AWA analysis is negative, the opposite of the actual difference. It is likely that in this region of high ozone precursor emissions chemical considerations dictate the ozone change and that changes in circulation play a minor role.

Applying the linear regression model based on geopotential height to a future climate gives a completely different picture than the one based on AWA (Fig. S3b). The model based on geopotential height predicts much larger future ozone changes, up to 10 ppb, over the southeastern US and ozone increases over southeastern Canada of up to 5 ppb. In the present climate, the linear regression model based on geopotential height is at least as good as that based on AWA. The difference in future projections can be explained by the fact that changes in AWA represent changes in the waviness of the circulation, but do not reflect changes
in the mean circulation; in contrast, the metric based on the geopotential height includes changes in the zonal mean and thus



reflects the change caused by warming in general, including a general northwards movement in the zonally averaged jet stream.

The sensitivity to changes in the AWA pattern (ppb/m$^2$) in the GCM2000 simulation and the sensitivity multiplied by the future change in AWA (GCM2100 minus GCM2000) is given in Fig. 11. Note, that the overall predicted ozone change is proportional to the sum of the latter metric summed over the study region (Fig. 11b, d, f). The CASTNET site HOW132 in the northeast US is positively sensitive to AWA changes right over the East coast, but negatively sensitive to changes further inland over the Midwest, and particularly in the upper Midwest. As a result of these competing influences the overall future change due to regional circulation changes is small (Fig. 11b). The effects of future AWA increases over both the East coast and over the Midwest largely cancel each other out. The sensitivity to changes in AWA in the mid-Atlantic region (e.g. at station PAR107) and in the southeast (e.g., at station GAS153) are opposite to those above. For these sites increases in the AWA in the central US increase the ozone anomaly (i.e., the sensitivity is positive) while AWA increases off the Eastern coast act to decrease the ozone anomaly (ie., the sensitivity is negative). At these sites the large positive change in AWA over the Midwest tends to dominate the future signal as the increased anticyclonic circulation over the Midwest results in more offshore flow.

## 4    Discussion and Conclusions

We use wave activity in a univariate linear regression model to quantitatively relate interannual variations in the large-scale flow over the U.S to interannual variations in ozone. At any point, the impact of AWA on ozone is measured through a projection of AWA onto the spatial structure of the sensitivity of ozone to AWA. Throughout much of the U.S. variations in wave activity explain 30-40% of the simulated and measured ozone variance (Fig. 5). While the explanatory value of AWA is not exceptionally high, in general the correlation between individual meteorological variables and ozone is generally not high (e.g., Sun et al., 2017; Kerr and Waugh, 2018). Here we find an interannual correlation between ozone and temperature is somewhat stronger than the relationship between AWA and ozone over the study region (mean of the former is 0.48 compared with mean of the latter is 0.24).

The variance explained by using geopotential height as the explanatory variable instead of AWA (Fig S3a) in the linear regression is roughly equivalent or somewhat more than that using wave activity. The advantage of using wave activity is its theoretical relationship to the large scale wave dynamics of the atmospheric circulation and its utility as a metric for blocking (Martineau et al., 2017). In addition, AWA is not explicitly sensitive to changes in zonal mean circulation characteristics, for example a general northward movement of the jet stream or a uniform increase in temperature (or equivalently geopotential thickness). Thus future changes in AWA give a fundamentally different picture of future ozone changes than changes in the geopotential height (Fig. 9 versus Fig. S3b). In particular, changes in AWA are related to changes in the local waviness of the flow.





As determined through the regression coefficients, there are two main centers of sensitivity for ozone variability over the Eastern US: one in the Midwest and one along the east coast (Fig. 9). The center along the east coast is likely controlled by the strength and position of the Atlantic anticyclone; the center over the Midwest is controlled by the strength and position of the AWA center over the western US. These centers control the flow of Gulf air into the eastern US and the transport of high
ozone air under anticyclonic conditions into the northeastern US. In most of the simulations analyzed here increased future AWA tends to increase ozone in the interior southeast and decrease it over the northeastern states.

The regression coefficients determining the ozone sensitivity to AWA within the study region are remarkably similar in the model simulations and in the measurement derived analysis, at least for the three representative points examined (Fig. S2).
Additionally, the ozone change in the model and measurements is nearly the same sign everywhere with the same approximate magnitude (Fig. 9) suggesting that the agreement in the model-derived and measurement-derived sensitivities is widespread. The exception to this agreement occurs at some of the CASTNET sites in the interior western US where the measurements and model predict a future ozone change of opposite sign. It is possible that local topographical features in this part of the US are important, features not captured by the model. The similarity between model and measurements suggests that at most
grid points the simulations do capture the conditions under which high (or low) ozone occur. Thus it is likely that model-measurement discrepancies in the ozone variance is due to differences in the AWA and not in the relation between AWA and ozone at a point.

Deficiencies in the simulation of the Atlantic anticyclone have been related to deficiencies in the simulation of ozone vari-
ability in the Northeast US. In the measurement derived analysis, the ozone variability associated with the MCA is large in the northeastern US; in the measurements the relation between AWA and ozone is strong, and changes in AWA can explain a significant fraction of the ozone variability (Fig. 5). In contrast, in the simulated MCA the relationship between ozone variability and AWA is weak in the northeastern US. Consistent with this the changes in AWA are not significantly related to changes in ozone in the northeastern US (Fig. 5). We attribute these differences between the simulation and the measurements
to differences in the simulation of the Atlantic anticyclone.

This has implications for future ozone extremes over the Northeast US. A rather robust feature of future climate change is the westward movement of the Atlantic anticyclone from its current position (Shaw and Voigt, 2015). In the GCM2000 simulation the Atlantic anticyclone is already situated considerably west of its climatological position. If the Atlantic anticyclone indeed
moves westward in the future, the MCA analysis of the GCM2000 simulation (Fig. 3d) (with its westward shift of the Atlantic anticyclone) might be a good representation of future ozone variability. If so we might expect that future high ozone pollution episodes associated with the Atlantic anticyclone over the Northeastern US will shift westward in the future with a consequent decrease in the variability of ozone over the northeast US.



The future simulations show considerable variability in their change in wave activity compared to present-day. This serves as an important reminder that circulation changes, even on a 25-year timescale show considerable variability. This long-timescale variability has been pointed with regards to climate (e.g., Deser et al., 2012) and with regards to long timescale changes in ozone (Hess and Zbinden, 2013; Lin et al., 2014; Hess et al., 2015; Barnes et al., 2016). Even on these fairly long 25 year

timescales the future change in ozone due to circulation variability is on the order of 2 ppb. Nevertheless, while some of the details differ, all future simulations examined with the CESM show a large enhancement in anticyclonic wave activity and geopotential height in the southwestern US. This enhancement of wave activity drives a significant fraction of the predicted future ozone change due to changes in wave activity. In all simulations this causes ozone increases through parts of the southeast US (Fig. 9). In the northeastern US future ozone change is generally small, ranging from about +1 ppb to -1 ppb depending on

the simulation (Fig. 9). Changes in AWA explain much of the future ozone change (GCM2100 minus GCM2000) outside the northeast US.

The difference in AWA between the GCM2000 simulation and the ERA-interim reanalysis is large (Fig. 6). The present day differences in AWA might be due to meteorological variability, or perhaps more likely, due to model bias. This implies that in

many locations ozone changes due to model bias (or variability) in AWA leads to ozone changes larger than those projected by future changes, with ozone increases of approximately 4-8 ppb in the southeastern US. The difference in AWA between the GCM2100 simulation and the GCM2000 simulation is similar to that between the GCM2000 simulation and the ERA-Interim reanalysis. Thus in many ways compared to the ERA-Interim reanalysis the GCM2000 simulation looks like what one would expect in a future climate. Over the western US the AWA anomaly gets larger as one goes from the ERA-interim reanalysis to

the GCM2000 simulation to the GCM2100 simulation. Similarly, off the east coast of the US the Atlantic anticyclone moves northwestward as one goes from the ERA-interim reanalysis to the GCM2000 simulation to the GCM2100 simulation (Fig. 4).

Given the fact that AWA in the GCM2000 simulation is in many ways more similar to that of the GCM2100 simulation than the ERA-interim analysis one might question whether the difference between the GCM2100 and GCM2000 simulations

provides an accurate metric for future change. The difference between the GCM2100 and GCM2000 simulations is consistent with the difference between the twenty-first century difference in the REFC2 ensemble members. The GCM2100 minus GCM2000 difference does not stand out compared to the REFC2 ensemble members (Fig. 8). Note also the GCM2000 simulation is similar to other future simulations in the westward movement of the Atlantic anticyclone and the strengthening of the anticyclone over the western U.S. (Shaw and Voigt, 2015). Thus the difference in AWA between the GCM2000 simulation and

the ERA-interim reanalysis (Fig. 6) or between the GCM2100 simulation and the ERA-interim reanalysis (Fig. S4) might be a better estimator of what to expect in the future. If the latter is the case increases in ozone of up to 15 ppb or decreases of up to 8 ppb might be expected due to changes in AWA (Fig. S4).

In conclusion, we show that wave activity, as a metric of large scale mid-latitude flow, provides a powerful tool to relate the

larger-scale tropospheric circulation to local surface ozone. In particular, we use changes in wave activity to better understand



the impact of model biases and future circulation changes on simulated surface ozone concentrations. A similar methodology could be expanded to other climate variables (e.g., temperature) or applied as a means of relating the larger scale flow field to surface ozone or temperature extremes. In all future simulations we find regionally robust summertime changes in wave activity over the western and southwestern US. It would be interesting to see if these future changes are robust across different

5    models and to further quantify their impact on predicted summertime climate change.



**Table 1.** CAM4-chem simulations used in this study.

| Simulation | GHG | Emissions | SST and sea ice | Meteorology |
|---|---|---|---|---|
| REFC2 (1960-2100) | RCP 6 | Anthropogenic and biomass burning from Assessment Report 5. Biogenic emissions from MEGAN. | Online | Online |
| GCM2000 (2000-2025, and use the last 20 years) | CO2 same as REFC2 for the period, other GHG from CMIP5 for 2000. | Constant value from REFC2 year 2000 | Online | Online |
| GCM2100 (2100-2125, and use the last 20 years) | CO2 same as REFC2 for the period, other GHG same as GCM2000. | Same as GCM2000 | Online | Online |

*Author contributions.* ST ran the GCM simulations. GC provided the expertise on local finite-amplitude wave activity. WS performed the analysis. WS and PH wrote the paper. All authors reviewed the paper and interpreted the data.

*Competing interests.* The authors declare that they have no conflict of interest.

*Acknowledgements.* This research was made possible by NSF award number 1608775. The CESM project is supported by the National
5   Science Foundation and the Office of Science (BER) of the U.S. Department of Energy. Computing resources were provided by the Climate Simulation Laboratory at NCAR's Computational and Information Systems Laboratory (CISL), sponsored by the National Science Foundation and other agencies.





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





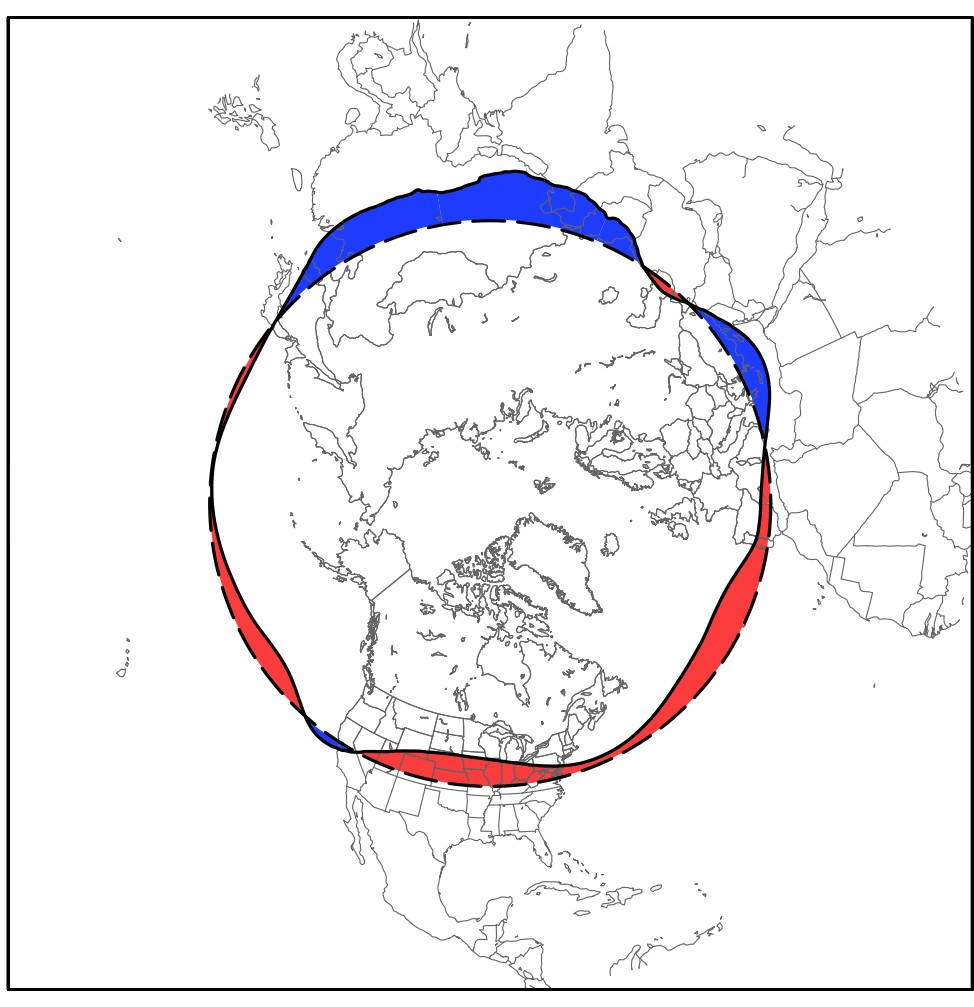

**Figure 1.** Anticyclonic local finite-amplitude wave activity (red) and cyclonic local finite-amplitude wave activity (blue) for the GCM2000 simulation based on the 25-year GCM2000 JJA average 5850 m geopotential height at 500 hPa (black contour). The dashed contour represents the equivalent latitude for this contour. At the equivalent latitude the magnitude of the local wave activity varies with longitude in proportion to the displacement of the black contour from the dashed contour. The total LWA$(\lambda, \phi_e)$ is equal to the cyclonic wave activity minus the anticyclonic wave activity.





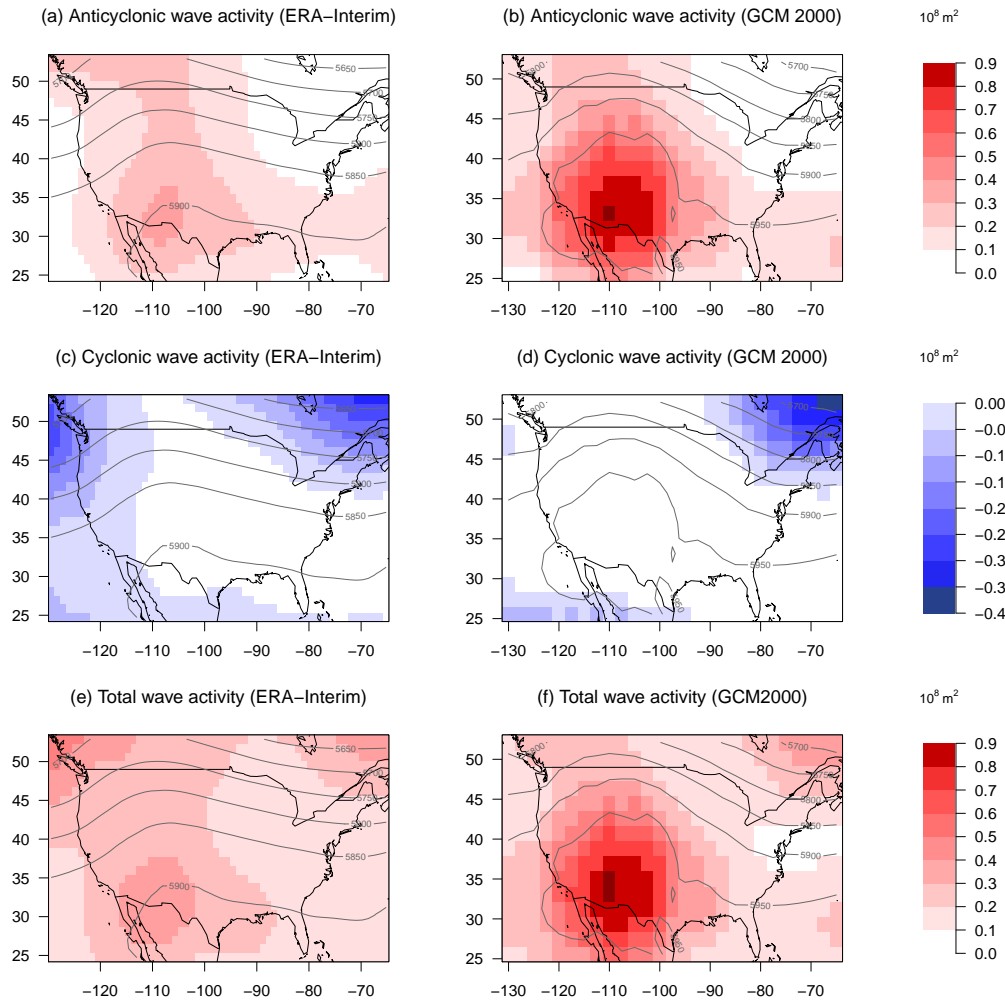

**Figure 2.** JJA climatology of wave activity in color shades ($10^8 m^2$). Anticyclonic local wave activity (a), cyclonic local wave activity (c) and the magnitude of total LWA (e) with 500 hPa geopotential height (contour, in m) for JJA 1979-2014 over the study region calculated from ERA-Interim reanalysis data. Climatology of anticyclonic local wave activity (b), cyclonic local wave activity (d) and the magnitude of total local wave activity (f) with 500 hPa geopotential height (contour, in m) for JJA 2006-2025 over the study region calculated from the GCM2000 simulation.





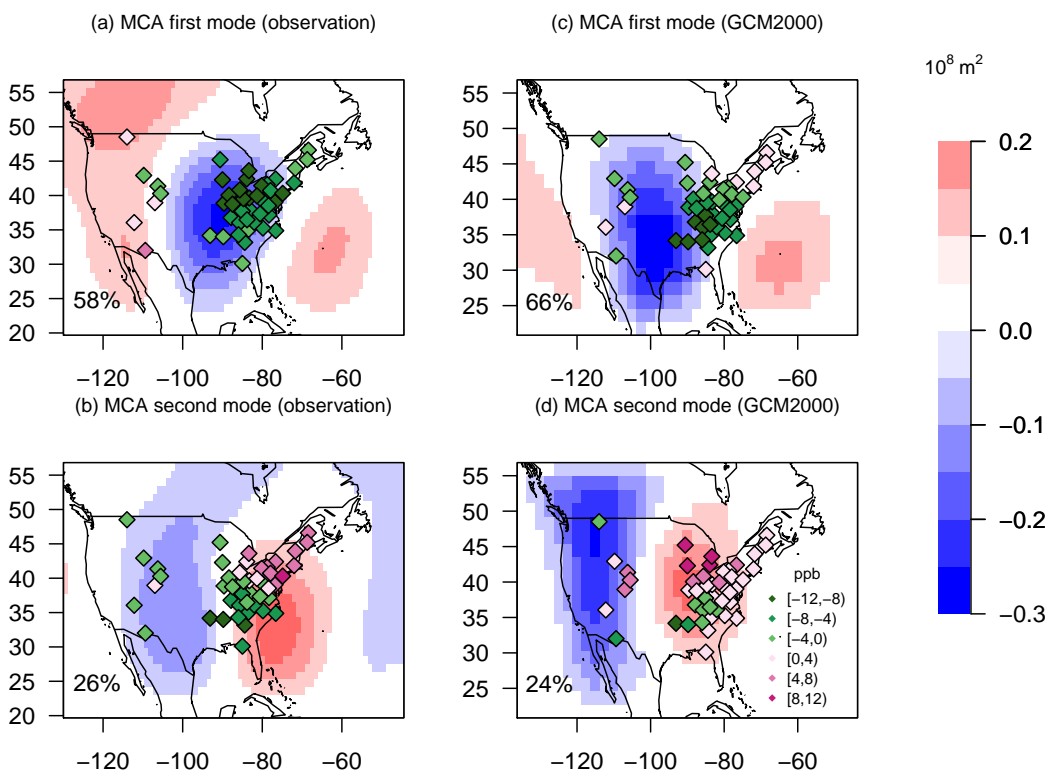

**Figure 3.** Spatial patterns of JJA AWA (color shades, in $10^8 m^2$) and MDA8 ozone (colored diamonds, in ppb) from the Maximum Covariance Analysis of AWA within the analysis domain and ozone at the CASTNET stations. The first mode (a) and second mode (b) from ERA-Interim meteorology and measured ozone at the CASTNET sites; (c),(d) as in (a) and (b) except from the GCM2000 simulation of AWA and ozone at the CASTNET sites. The percent variance explained by each mode is given in the lower-left corner of each panel.





**Figure 4.** JJA 20-year average streamfunction ($10^6 m^2 s^{-1}$) at 850 hpa for ERA interim (shade), GCM2000 simulation (black contour) and GCM2100 simulation (grey contour).





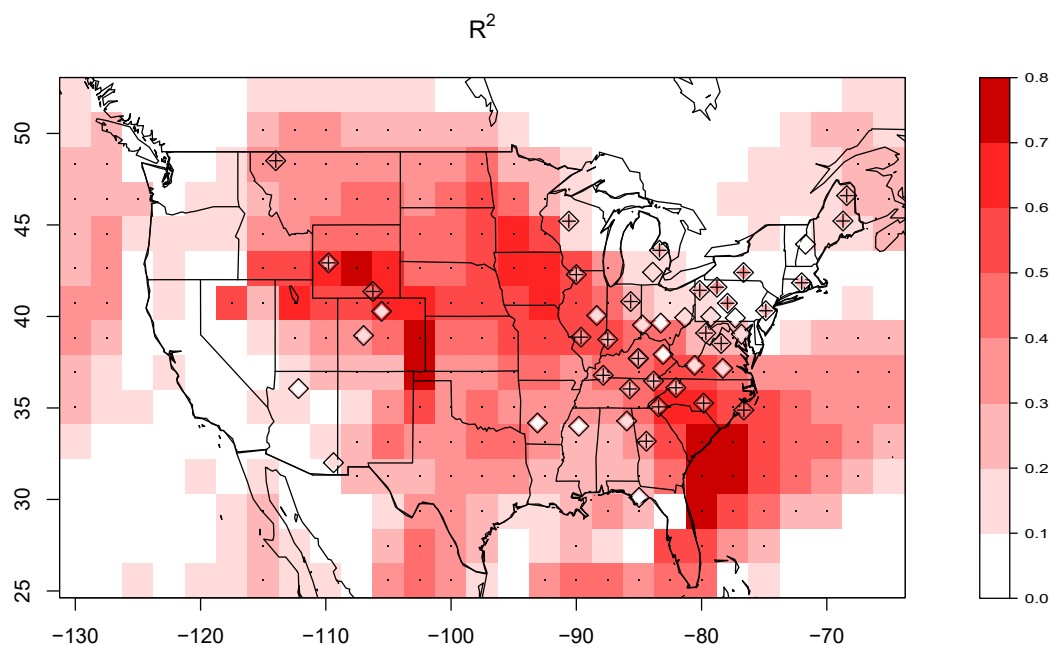

**Figure 5.** Interannual variance of MDA8 ozone explained ($R^2$) by the linear regression model (Eq. 5) using the AWA projection value (Eq. 4) as the explanatory variable, with simulated variance in shades (as derived from simulated ozone and AWA in the GCM2000 simulation) and measured variance in diamonds (as derived from measured ozone at CASTNET sites and the ERA-interim meteorology). Plus signs and stippling represent where $R^2$ is significant (at the 5% significance level) at CASTNET sites and model grids, respectively.





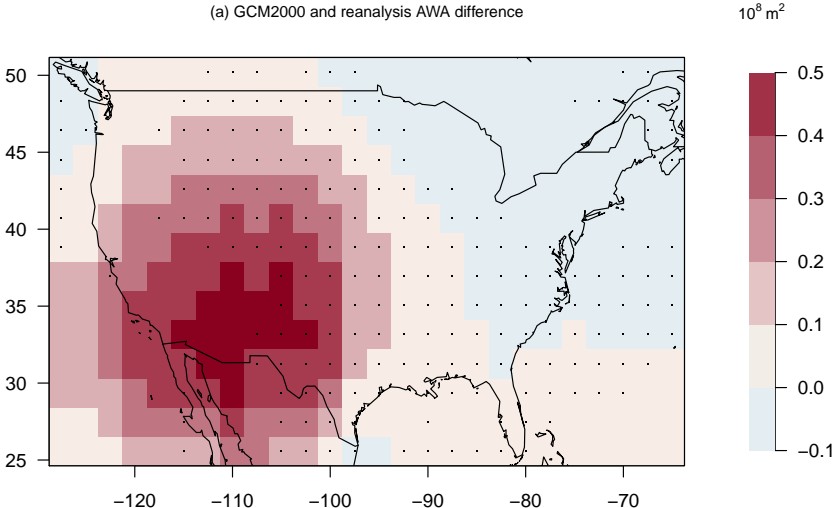

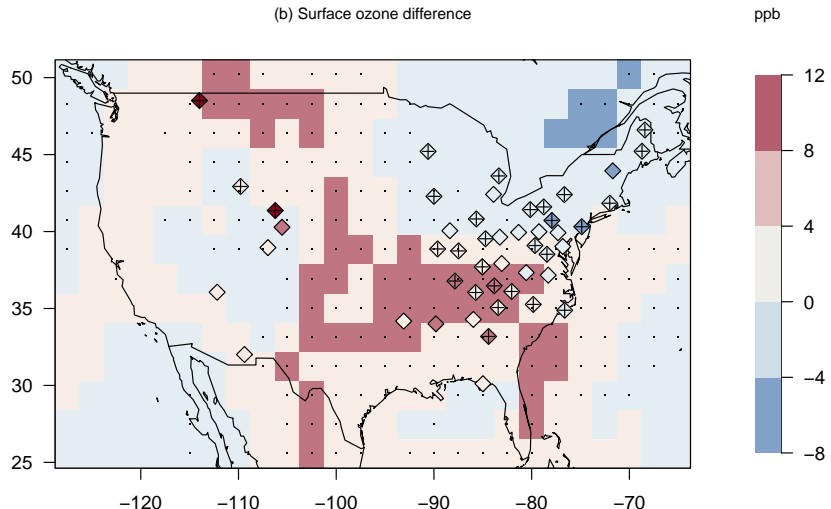

**Figure 6.** (a) JJA difference in AWA between the GCM2000 simulation (2006-2025) and the ERA Interim analysis (1995-2014) ($10^8 m^2$). (b) Change in MDA8 ozone from the linear regression model derived from the GCM2000 simulation (shaded) or the measurements (diamonds) using the difference in AWA calculated from GCM2000 simulation and ERA-Interim analysis as the explanatory variable. Plus signs and stippling represent the ozone difference is significant (at the 5% significance level) at CASTNET sites and model grids, respectively.



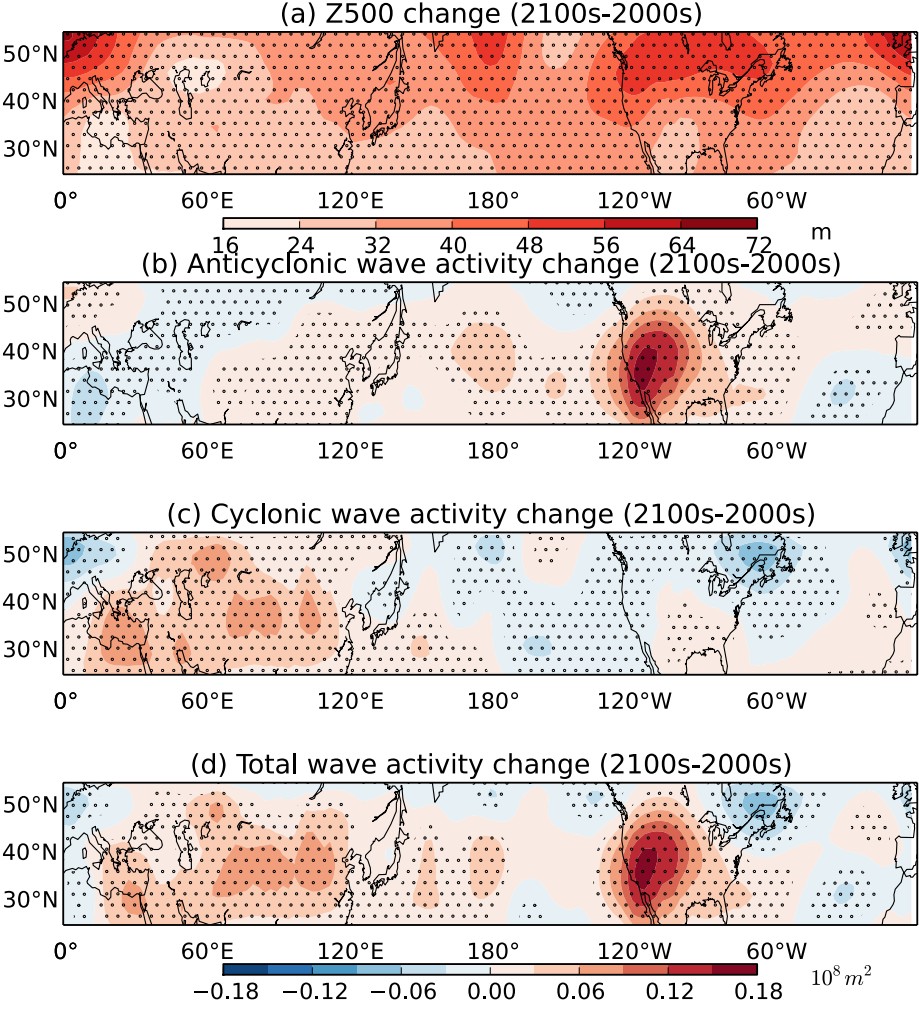

**Figure 7.** JJA simulated difference between the GCM2100 and the GCM2000 for 500 hPa geopotential height (m) (a) anticyclonic wave activity ($10^8 m^2$) (b) cyclonic wave activity ($10^8 m^2$) (c) and total wave activity ($10^8 m^2$) (d) between 24N and 53N. Note that in the northern hemisphere the cyclonic wave activity and total wave activity are negative. Here the change in magnitude is presented. Stippling represents where the change is significant at 5% level using a Student t test.



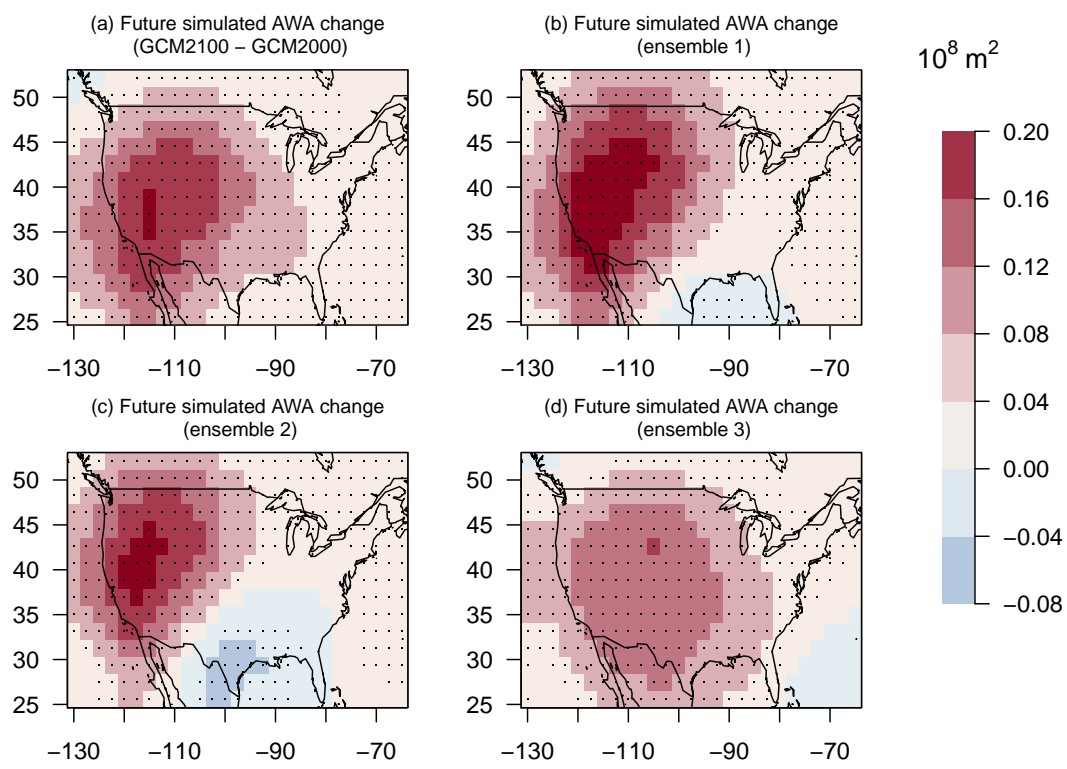

**Figure 8.** JJA difference ($10^8 m^2$) in AWA between the GCM2100 and the GCM2000 simulation (a) and between the three ensemble members of the REFC2 simulation between 2090-2099 and 2000-2009 (b-d). Stippling represents where the change is significant at 5% level using a Student t test.



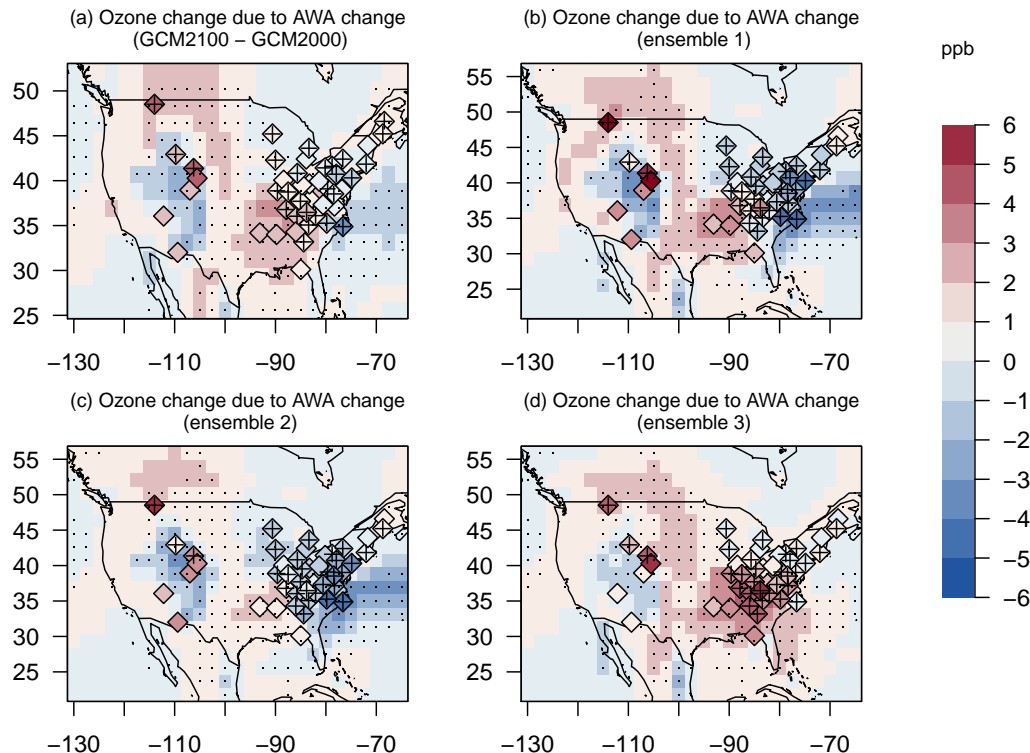

**Figure 9.** Change in MDA8 ozone from the linear regression model derived from the GCM2000 simulation (shaded) or the measurements (diamonds) using the difference in AWA as the explanatory variable. Differences in AWA are calculated between the GCM2100 simulation and GCM2000 simulation (a) between the three ensemble members of the REFC2 simulation between 2090-2099 and 2000-2009 (b-d). Plus signs and stippling represent where the future change of AWA is significant (at the 5% significance level) at CASTNET sites and model grids, respectively.




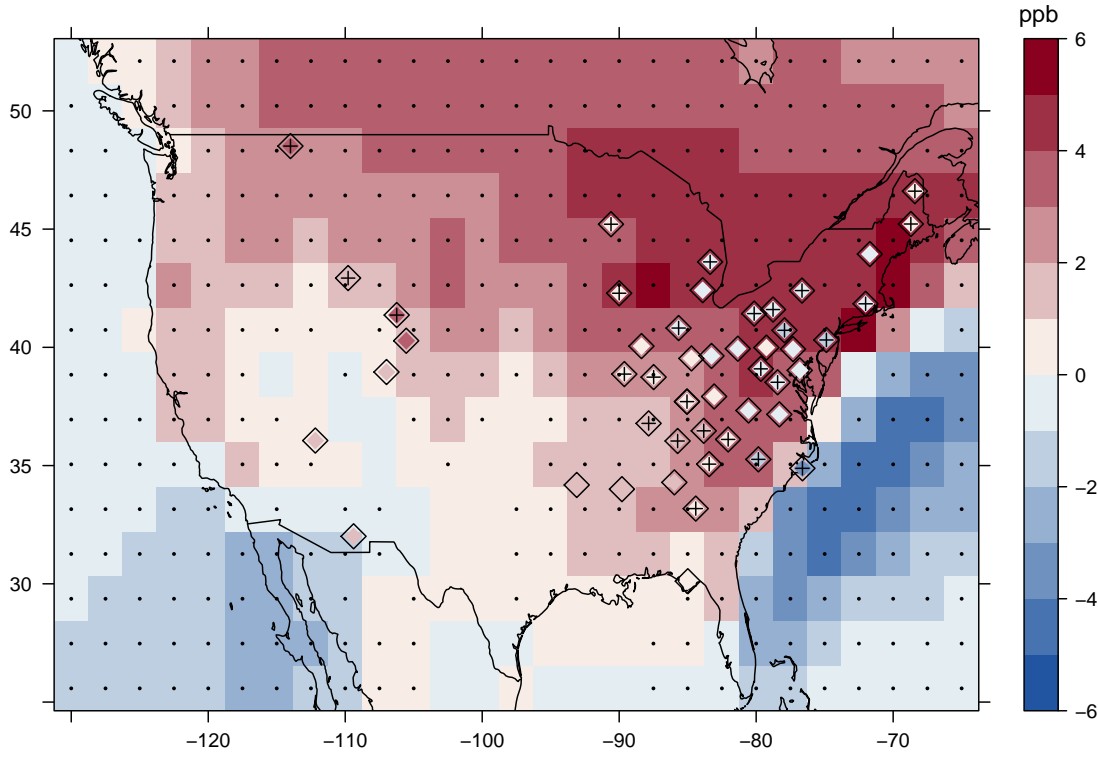

**Figure 10.** JJA change in MDA8 ozone between the GCM2100 simulation and the GCM2000 simulation (ppb, shaded). Diamonds give the change in MDA8 ozone in the linear regression model derived from the measurements using the difference in AWA between GCM2100 and GCM2000 as the explanatory variable.





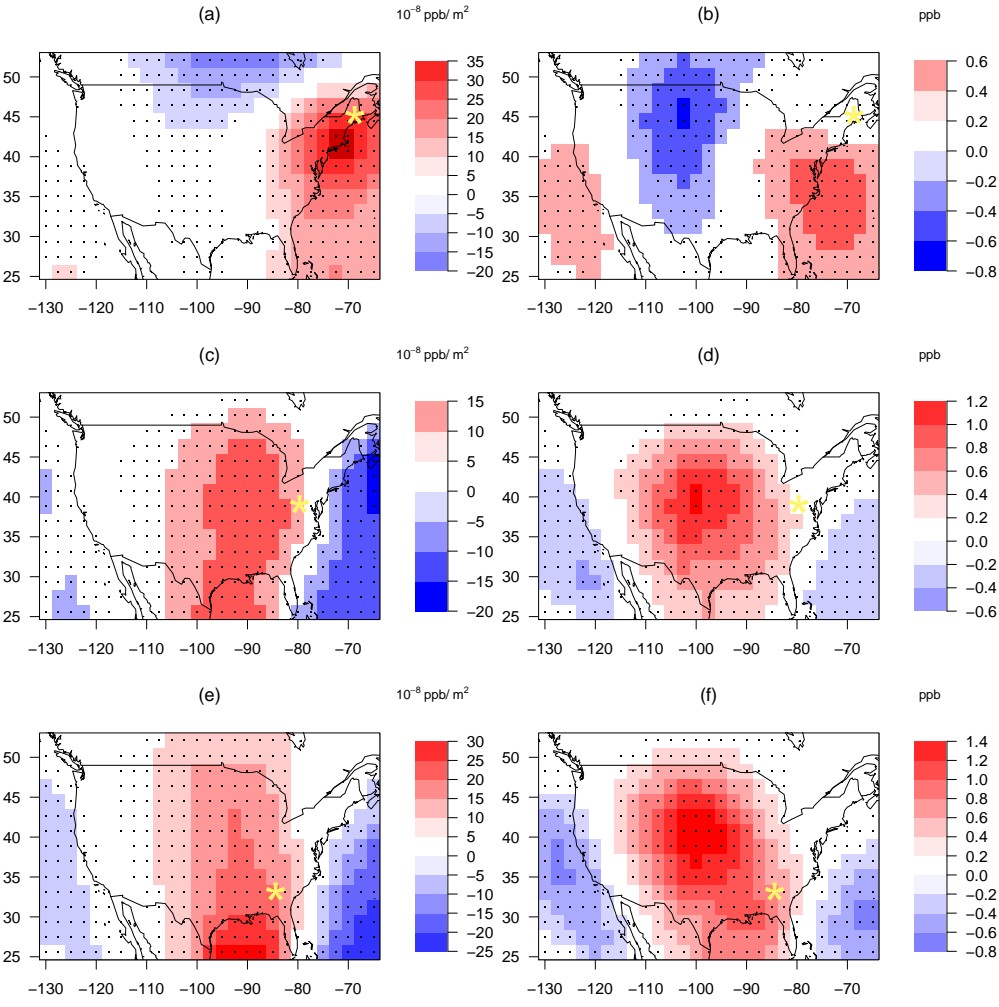

**Figure 11.** Regression coefficients between MDA8 ozone during JJA at representative sites (star sign) and AWA in the study region (a, c, e). Regression coefficients multiplied by the change in AWA between the GCM2100 simulation and the GCM2000 simulation (b, d, f). The predicted change in ozone from the linear regression model is proportional to the sum of all points over the domain (Eq. 5). Stippling represents where the regression coefficient is significant at the 5% significance level.