# Peer review of "How waviness in the circulation changes surface ozone: a viewpoint using local finite-amplitude wave activity"

_Atmospheric Chemistry and Physics, 2019_

## Referee Comment (RC1) · Anonymous Referee #1 · 11 Jun 2019

This work by Sun et al. applies an advanced metric of wave activity to the study of air pollution. The relationships generated from observed and modeled ozone variations are used with projected changes in wave activity to quantify the impacts of climate change and also suggest the role of meteorological shortcomings in the persistent high bias in surface ozone. The manuscript is novel and will be of interest to a dedicated set of researchers studying this issue. I recommend publication after the authors address the following, mostly minor, comments.

**General Comments**

- I am somewhat familiar with LWA, but have always struggled to understand the

mapping from equivalent latitude back to the conventional geographic latitude. A brief discussion of this issue would be beneficial, especially for the unaccustomed reader.

- The ozone dataset is normalized to have unit variance. Is the local variance introduced back into the analysis of the projection value of the ozone influence from AWA?

- I understand the need to look at a broader time period to analyze the ozone-AWA relationship, but I do not follow the logic presented in equations 5 and 6. Can $\beta$ be interpreted as the importance of the pattern contribution to the seasonal value? And then $\alpha$ is a sort of baseline amount? It seems to me that $\beta$ will change with emissions and climate, and also be intricately linked to AWA. It is stated on Page 12 that $\beta$ doesn't change, but shouldn't it? If more days are under larger AWA or the magnitude changes, I would expect the contribution to seasonal averages to change.

- Are the quantities in Figure 2 estimated from monthly mean geopotential heights or the average of daily AWA/LWA? Should there be a difference?

- Is AWA normalized to unit variance for the MCA analysis? It strikes me that the variables should have comparable variance in order to prevent one from dominating the results.

- I have to ask about the reliability of the projections given an ensemble of three and relatively short analysis record. There is some discussion, but I think a bit more is warranted.

- What impact does the spatial and temporal resolution of the geopotential height fields have on the estimate of AWA? I assume additional structure is available with higher spatial resolutions. The resolution used here in the reanalysis and

climate model output is rather course. Would a higher spatial resolution product improve the relationships?

**Specific Comments**

- Page 2, Lines 7, 11, 12 - There are some citation formatting issues here.

- Equation 2 - I am confused how both the cyclonic and anticyclonic LWA integrate inclusively to $\phi_e$

- Page 5, Line 23 - Stating "scenario" and "pathway" is redundant.

- Table 1 - The SST/Sea Ice and Meteorology columns are unnecessary if they're all the same value (online)

- Page 5, Lines 32, 33, 34 - Subscript missing in $CH_4$ and $CO_2$.

- Equation 5 - Is $O_{3,i_0,j_0}(t)$ the seasonal (JJA) average of MDA8?

- Page 8, Line 29 - A period has gone astray.

- Page 9, Lines 1-7 - Does the interpretation of Shen et al. (2015) for their first two EOF modes match the physical explanation for the first two MCA patterns presented here?

- Page 9, Lines 1-3 - Is there any reason to prefer this method over the EOF analysis of 500hPa heights?

- Page 9, Line 19 - "less also" should probably be just "less"

- Page 9, Lines 18-19 - This is too be expected, right? The southeastern US flow from the Gulf is more mesoscale and likely poorly resolved transport in a global model at this resolution.

---

## Referee Comment (RC2) · Anonymous Referee #2 · 17 Jun 2019

This study explores the relationship of wave activity and surface ozone in the United States. In general, I think this manuscript is well structured and the topic is suitable for ACP. But the physical mechanism linking the wave activity with surface ozone is not clear. Also I feel this study seems to over interpret the role of wave activity (or asymmetric regional circulation), and I am not very convinced it is a useful metric that can explain much of the future changes of surface ozone. I think the authors need to address these concerns before ACP accepts this paper.

Major comments.

The physical mechanism linking AWA and ozone is unclear. If AWA affects ozone

through altering the transport, why not directly use the wind or vorticity? If the authors think the asymmetric regional circulation can add more information, they should show what is lacking if they only use the general circulation patterns.

As mentioned by this study, AWA is reflective of asymmetric regional circulation changes with respect to the zonal mean, which is only part of the weather variability that influences ozone air quality. Other weather factors like temperature and general circulation patterns should play even more important roles in modulating ozone variability. If we say weather can affect ozone in the thermal (e.g. temperature and relative humidity) or dynamical pathways (transport and ventilation), then AWA is only part of the dynamical one. So it is not surprising that it can explain only a small fraction of ozone variance (Figure 5). So this study shouldn't over-interpret the importance of AWA. I think the authors need to more accurately describe why AWA is useful.

Given this, we shouldn't expect AWA to be able to capture the future ozone changes. The authors claim that in many locations the linear regression model using AWA as the predictor can explain the magnitude of the simulated ozone changes (P12L24-25), but the spatial patterns as shown in Figure 9 and 10 are quite different. Even the magnitude of predicted ozone changes from these two methods may match in some gridboxes, this may be just a coincidence and it can't support that AWA can be used to predict (or predict much of) future ozone changes.

Also, weather variables are not independent. Or we can say that the slopes of ozone with AWA should also include effects of other meteorological variables. So will this affect the conclusion that AWA is reflective of asymmetric regional circulation changes with respect to the zonal mean? Does AWA really explain some part of ozone variances that can not be explained by other variables?

Equation 4 and 5. Will the domain size affect $pi_{0,j_0}$? Shouldn't the domain also move with the location of gridbox $(i_0,j_0)$?

Minor comments. P1L5. It is not surprising to find high fraction of explained covariance

if using MCA. The author should further give the numbers of explained variance in MDA8 ozone.

P1L22. Please specify the timeframe of 'future' change.

P2L3-4. Please make it clear that the stagnant weather definition used by Kerr and Waugh (2018) is from Wang et al. (1998). Ozone may still be strongly dependent on the stagnant weather, but the definition of stagnant weather from Wang 1998 may not be appropriate. This paper is cited in Kerr and Waugh (2018).

P2L30. There are already some studies that have tried to explain the relationship of wave activity and surface ozone air quality (e.g., Shen et al. (2017) and maybe some papers cited therein, https://doi.org/10.1073/pnas.1610708114 ).

P3L10. Turner et al. (2013) didn't use real observations, so this may explain why they found weak relationships. Many studies that use real observations indeed found strong correlations between cyclone frequency and high-ozone events. I think the authors should cite these observations based studies rather than Turner et al. (2013).

P5L15. The authors should give a brief summary of the ozone chemistry used in the model.

P5L21. Are these three ensemble simulations the same? It is not clear to readers.

P8L31. This study should also report the explained variance of MDA8 ozone.

P9L22. Seems Figure 4 can be moved to the supplement.

P10L10. It seems the AWA can only explain a small fraction of ozone variance.

P12L24. I don't see that the pattern in Figure 9 can match that in Figure 10.

---

## Author Comment (AC1) · 14 Aug 2019

We would like to thank the reviewer for their time and effort in reviewing the manuscript. Their comments have improved the manuscript for which the authors are greatful. Below are our point-to-point responses to the comments. For clarity, we colored our responses in blue.

• I am somewhat familiar with LWA, but have always struggled to understand the mapping from equivalent latitude back to the conventional geographic latitude. A brief discussion of this issue would be beneficial, especially for the unaccustomed Reader.

We're not sure how to construct a map from equivalent latitude back to the geographic latitude. Perhaps a better way to think about it is that for every geographic latitude there is a contour of potential vorticity (or in our case geopotential height) for which that latitude is the equivalent latitude. As stated in the text the area within the equivalent latitude circle is equal to the area within the q=Q contour. So for any latitude we can find a contour for which the area within the contour is the same as the area within that latitude circle. This is also shown in Figure 1. (No change to text).

• The ozone dataset is normalized to have unit variance. Is the local variance introduced back into the analysis of the projection value of the ozone influence from AWA?

This is a good point. We did not normalize the time series but just deseasonalized it. We state now on Page 6: "The linear relationship between changes (in time) of deseasonalized ozone at a point $(i0, j0)$ and changes in deseasonalized wave activity at another point $(i, j)$, can be simply expressed as the slope of ozone with respect to wave activity (ppb/m2)$(Si0,j0(i,j))$."

• I understand the need to look at a broader time period to analyze the ozone-AWA relationship, but I do not follow the logic presented in equations 5 and 6. Can beta be interpreted as the importance of the pattern contribution to the seasonal value? And then alpha is a sort of baseline amount? It seems to me that beta will change with emissions and climate, and also be intricately linked to AWA. It is stated on Page 12 that beta doesn't change, but shouldn't it? If more days are under larger AWA or the magnitude changes, I would expect the contribution to seasonal averages to Change.

Yes, beta is the contribution of the pattern to the ozone concentration and alpha is the background value. To clarify this we have amended the text to read:

"Therefore, to correct for this we build a linear regression model where we relate O3i0,j0 (t) to pi0,j0 (t) through the linear regression coefficients alphai0,j0 and betai0,j0, where beta is a measure of the overall sensitivity of ozone to AWA and alpha gives the ozone background concentration:"

Alpha and Beta might change with changes in emissions or the emissions distribution, but here we are interested in the emissions independent change (e.g., the change due to changes in the circulation). We expect the sensitivity of future ozone to future wave activity (ppb/m2) (assuming no changes in emissions) to be the same as the present sensitivity. In other words, the same pattern of wave activity would be expected to change ozone to the same extent in the future as at present. To clarify this we have added: "The fact that Beta does not significantly change in the future is confirmed by an analysis of the GCM2100 simulation (see section 3.3.2)."

The background value of ozone (reflected in alpha) might change in the future due to changes in temperature, but changes in alpha are not represented in equation 6.

• Are the quantities in Figure 2 estimated from monthly mean geopotential heights or the average of daily AWA/LWA? Should there be a difference?

We use the average of daily AWA. We have clarified this in the caption. There should be a difference. Use of monthly mean geopotential heights would result in an AWA with smaller magnitude.

• Is AWA normalized to unit variance for the MCA analysis? It strikes me that the variables should have comparable variance in order to prevent one from dominating the results.

No. MCA does not require the scaling of the input. Normalization does not change the result.

• I have to ask about the reliability of the projections given an ensemble of three and relatively short analysis record. There is some discussion, but I think a bit more is warranted.

As the reviewer recognizes there is considerable variability in the ensemble means. In the present climate we examined a 20-year time period due to limitations in the data record. For consistency we also used a 20-year period averaging period in the ECMWF data and in the two constant climate simulations (GCM2000 and GCM2100). In the ensemble simulations we examined a 10-year period as these simulations are non-stationary and we wanted to examine differences representative of the 100 year timespan between 2100 and 2000. We point out the variability due to this short averaging period in reference to Fig. 8 (with implications for the variability in Fig. 9) and in the conclusion. We point out the importance of this variability in interpreting long-term trends. It is unclear to us how much more discussion of this is warranted in the paper.

• What impact does the spatial and temporal resolution of the geopotential height fields have on the estimate of AWA? I assume additional structure is available with higher spatial resolutions. The resolution used here in the reanalysis and climate model output is rather course. Would a higher spatial resolution product improve the relationships?

For Fig. 2, the reanalysis and the model used geopotential height at different resolutions to calculate AWA. The reanalysis has a spatial resolution of 1.125*1.125 and the model has the resolution as 2.5*1.9. The AWA patterns are similar at these different spatial resolutions so we doubt the higher spatial resolution would improve the estimate of AWA. Furthermore, one of the points of using an analysis based on AWA is it captures the larger spatial scale aspects of the circulation (aspects where we might expect to capture future changes) as opposed to the fine details. Thus we do not feel that higher resolution products would improve the relationships.

Specific Comments

• Page 2, Lines 7, 11, 12 - There are some citation formatting issues here.

Changed in text.

• Equation 2 - I am confused how both the cyclonic and anticyclonic LWA integrate inclusively to equivalent latitude.

The cyclonic part of the wave activity is that part south of the equivalent latitude (AC < 0) while the anticyclonic part is to the north (AA > 0). Note that the total LWA is defined by AC - AA (Equation 3) so these two parts don't cancel each other.

• Page 5, Line 23 - Stating "scenario" and "pathway" is redundant.

'Pathway' is removed in the text.

• Table 1 - The SST/Sea Ice and Meteorology columns are unnecessary if they're all the same value (online)

These columns are removed in the text.

• Page 5, Lines 32, 33, 34 - Subscript missing in CH4 and CO2.

Changed in text.

• Equation 5 - Is O3;i0;j0(t) the seasonal (JJA) average of MDA8?

It is the JJA average of deseasonalized MDA8.

• Page 8, Line 29 - A period has gone astray.

Changed in text.

• Page 9, Lines 1-7 - Does the interpretation of Shen et al. (2015) for their first
two EOF modes match the physical explanation for the first two MCA patterns
presented here?

The MCA patterns we find are consistent with those of Shen et al. (2015). Consequently, we
assume the MCA patterns found in this study are consistent with the physical explanation in
Shen et al. (2015)

• Page 9, Lines 1-3 - Is there any reason to prefer this method over the EOF analysis
of 500hPa heights?

MCA is a more elegant way to find the primary modes of variability between two fields than
using the two-step procedure of performing an EOF on one field and then correlating with
another field. It also offers a more concise interpretation of how the variability of the fields are
related.

• Page 9, Line 19 - "less also" should probably be just "less"

Changed in text.

• Page 9, Lines 18-19 - This is too be expected, right? The southeastern US flow
from the Gulf is more mesoscale and likely poorly resolved transport in a global
model at this resolution.

We would expect that a model at this resolution would be able to simulate a large scale feature
such as the North Atlantic subtropical anticyclone. While we might expect a global model not to
correctly simulate details of the Great Plain low level jet, it should be able to simulate the overall
flow.

---

## Author Comment (AC2) · 15 Aug 2019

We would like to thank the reviewer for their time and effort in reviewing the manuscript. Their comments have improved the manuscript for which the authors are greatful. Below are our point-to-point responses to the comments. For clarity, we colored our responses in blue.

Major comments.

The physical mechanism linking AWA and ozone is unclear. If AWA affects ozone through altering the transport, why not directly use the wind or vorticity? If the authors think the asymmetric regional circulation can add more information, they should show what is lacking if they only use the general circulation patterns.

As stated in the introduction AWA is a metric to diagnose general circulation patterns in the large-scale flow: "A diagnosis of LWA also provides a metric for the occurrence of blocking events, events associated with anomalous or extreme mid-latitude weather events such as heat waves (Chen et al., 2015; Martineau et al., 2017) which have been associated with surface ozone extremes (e.g., Sun et al., 2017; Meehl et al., 2018; Phalitnonkiat et al., 2018)." It is important to examine large scale features as "the impact of climate change on large scale features of the circulation is likely more robust than that on smaller scales and in some cases, large-scale changes in circulation can be inferred from general theoretical arguments". We are likely to be less confident in future changes in smaller scale features. Wind fields and vorticity fields are expected to be noisy. Moreover, it is unclear exactly where and on what level we should analyze these fields (e.g., at the surface?). Local fields are likely to be influenced by diurnal circulations, such as mountain-valley, or land-sea breezes making them harder to interpret. Finally, LWA is closely tied to flow dynamics. "Since blocking events are related to the flux and convergence of LWA, the processes that control LWA may provide clues to how blocking will change in the future (Nakamura and Huang, 2018)." Thus LWA, argued in the introduction, is potentially a good candidate for relating surface ozone to characteristics of the general circulation. Local wind fields or vorticity fields do not have this advantage.

We have emphasized in the text the importance of more local fields in ozone prediction. However, we believe AWA as a metric of the larger scale circulation also provides insight. We have added some sentences at the beginning of the introduction summarizing this (see comment below).

As mentioned by this study, AWA is reflective of asymmetric regional circulation changes with respect to the zonal mean, which is only part of the weather variability that influences ozone air quality. Other weather factors like temperature and general circulation patterns should play even more important roles in modulating ozone variability. If we say weather can affect ozone in the thermal (e.g. temperature and relative humidity) or dynamical pathways (transport and ventilation), then AWA is only part of the dynamical one. So it is not surprising that it can explain only a small fraction of ozone variance (Figure 5). So this study shouldn't over-interpret the importance of AWA. I think the authors need to more accurately describe why AWA is useful.

We have added the following to the last paragraph in the introduction to try to summarize the arguments that we outlined earlier. In particular we are interested in regional circulation changes which makes LWA versus more localized regional variables a good choice:

"Here we focus on the extent that LWA is related to surface ozone and use this to predict the impact of circulation changes on ozone in the present and future climates. The advantages of using LWA are that it: i) provides a concise metric of regional circulation and its changes, ii) provides a metric for anomalous mid-latitude weather events which have been associated with high surface ozone concentrations, iii) is fundamentally related to the large-scale flow field through finite amplitude wave activity. Our emphasis on LWA does not preclude the importance of local meteorological effects on ozone such as the impact of local cloudiness, temperature, boundary layer ventilation and wind direction. Indeed, as discussed below, local temperature has generally more predictive power than LWA on ozone. However, while local changes in temperature, for example, are important, and indeed are related to the circulation changes characterized by LWA, it is difficult to diagnose changes in circulation from local temperature alone. Moreover, the prediction of future regional changes in circulation as characterized by LWA are likely more robust than future predictions of more local changes in temperature."

We agree that temperature might be the single variable that explains most of the ozone variance. However, hopefully the addition of the comments given above summarizes why we chose to use AWA. In fact, if we regress changes in ozone with temperature (as we did with the projection of AWA) and compute the R squared for each of the gridbox, we can compare the ability of temperature versus AWA to explain ozone changes. Indeed, Fig. 1 shows the histogram of difference in R squared between using AWA and temperature. It suggests that on average using AWA explains 20% less total ozone interannual variance than using temperature. However, using temperature does not show us how future changes in circulation impact ozone as the temperature change is impacted by more than simply changes in the circulation.

[Figure]

Figure 1. Histogram of difference in R squared between two linear models using AWA projection value and temperature projection value respectively.

Given this, we shouldn't expect AWA to be able to capture the future ozone changes. The authors claim that in many locations the linear regression model using AWA as the predictor can explain the magnitude of the simulated ozone changes (P12L24-25), but the spatial patterns as shown in Figure 9 and 10 are quite different. Even the magnitude of predicted ozone changes from these two methods may match in some gridboxes, this may be just a coincidence and it can't support that AWA can be used to predict (or predict much of) future ozone changes.

We don't claim that changes in AWA can capture the future ozone changes, only that it can be used to show the impact of future changes in circulation on ozone. We have added the following sentence: "Comparing the GCM2100 and the GCM2000 simulations, future changes in ozone over land range from approximately -1 ppb to 5 ppb. It is clear that overall, changes of circulation, as defined through changes in AWA, can not explain future ozone changes."

However, there are some locations where the change of ozone due to change in AWA is consistent with the future ozone change. We claim in the paper: " In many locations the change in ozone predicted from the change in wave activity between the GCM2100 and GCM2000 simulations is consistent with the actual ozone change." It is true that this might be due to

coincidence, but we are only claiming consistency here. We are not claiming that AWA can be used to predict (or predict much of) future ozone changes, but that it can be used to predict the result of future circulation changes on ozone.

Also, weather variables are not independent. Or we can say that the slopes of ozone with AWA should also include effects of other meteorological variables. So will this affect the conclusion that AWA is reflective of asymmetric regional circulation changes with respect to the zonal mean? Does AWA really explain some part of ozone variances that can not be explained by other variables?

AWA reflects the overall pattern. Changes in AWA does include effects of other meteorological variables which directly control ozone. The part of the ozone variance that it explains can likely be captured by other variables. Nevertheless, it is a convenient metric of the impact of regional circulation changes on ozone. We have added the following couple sentences to the conclusion: "Nevertheless, as a metric of changes in the regional circulation, AWA explains a significant fraction of interannual ozone change. Changes in AWA, of course, impact local variables that directly control ozone (e.g., temperature, boundary layer venting etc.)."

Equation 4 and 5. Will the domain size affect $pi_{0,j_0}$? Shouldn't the domain also move with the location of gridbox $(i_0,j_0)$?

This is a good point. If the domain moves with the location of gridbox, more of ozone's variability especially on the west coast could be explained. Moving the domain with the gridbox is a good idea, but since most of this study concentrates on the Eastern part of the country it is unlikely to impact the major conclusions.

Minor comments.

P1L5. It is not surprising to find high fraction of explained covariance if using MCA. The author should further give the numbers of explained variance in MDA8 ozone.

The first mode explains 29% of the total MDA8 ozone variance and the second mode explains 14% of the total MDA8 ozone variance. These fractions are now given in the text.

P1L22. Please specify the timeframe of 'future' change.

Changed in text.

P2L3-4. Please make it clear that the stagnant weather definition used by Kerr and Waugh (2018) is from Wang et al. (1998). Ozone may still be strongly dependent on the stagnant weather, but the definition of stagnant weather from Wang 1998 may not be appropriate. This paper is cited in Kerr and Waugh (2018).

Kerr and Waugh (2018) used Air Stagnation Index (ASI) as detailed in Horton et al. (2012). Quoting from Kerr and Waugh (2018): "The definition of the ASI used in this study and described in Horton et al (2012, 2014) slightly differs from the definition used by the National Centers for Environmental Information (NCEI, formerly the National Climatic Data Center), detailed in Wang and Angell (1999) and Korshover and Angell (1982)."

P2L30. There are already some studies that have tried to explain the relationship of wave activity and surface ozone air quality (e.g., Shen et al. (2017) and maybe some papers cited therein, https://doi.org/10.1073/pnas.1610708114 ).

Thanks for pointing out this reference. We have added in the introduction: "Shen et al. (2017) note the connection between the eastward-propagating flux in wave activity associated with the Pacific extreme pattern and increased surface pressure, reduced precipitation, warmer temperatures, more frequent heat waves and enhanced ozone over the eastern U.S."

We have not found papers therein that point out the connection between wave activity and ozone.

P3L10. Turner et al. (2013) didn't use real observations, so this may explain why they found weak relationships. Many studies that use real observations indeed found strong correlations between cyclone frequency and high-ozone events. I think the authors should cite these observations based studies rather than Turner et al. (2013).

We deleted the sentence that suggests the weak relationship between cyclone frequency and high ozone events. We have also included the reference to Leibensperger et al (2008) which is more observationally based.

P5L15. The authors should give a brief summary of the ozone chemistry used in the Model.

Ozone chemistry is described in Lamarque et al. (2012) in detail. We have included some additional text summarizing the chemistry.

P5L21. Are these three ensemble simulations the same? It is not clear to readers.

They are different in their initial conditions. Changed in text.

P8L31. This study should also report the explained variance of MDA8 ozone.

The explained variance of MDA8 ozone is added in the text.

P9L22. Seems Figure 4 can be moved to the supplement.

This figure shows the GCM2000 simulates the Bermuda high's position more to the west compared with the reanalysis and explains the difference in MCA's 2nd mode position between GCM2000 and the reanalysis. We prefer to leave it in the text.

P10L10. It seems the AWA can only explain a small fraction of ozone variance.

The AWA explains a statistically significant fraction of the ozone variance.

P12L24. I don't see that the pattern in Figure 9 can match that in Figure 10.

Please see answer to 'major comments' above.

References

Horton, Daniel E., and Noah S. Diffenbaugh. "Response of air stagnation frequency to anthropogenically enhanced radiative forcing." *Environmental Research Letters* 7.4 (2012): 044034.

Lamarque, J-F., et al. "CAM-chem: Description and evaluation of interactive atmospheric chemistry in the Community Earth System Model." *Geoscientific Model Development* 5.2 (2012): 369.

---

## Author Comment (AC3) · 15 Aug 2019

The comment was uploaded in the form of a supplement:
https://www.atmos-chem-phys-discuss.net/acp-2019-383/acp-2019-383-AC3-
supplement.pdf